# The Thermal Ablation with MRgFUS: From Physics to Oncological Applications

**DOI:** 10.3390/cancers17010036

**Published:** 2024-12-26

**Authors:** Mario Leporace, Ferdinando F. Calabria, Roberto Siciliano, Carlo Capalbo, Dimitrios K. Filippiadis, Roberto Iezzi

**Affiliations:** 1Department of Nuclear Medicine and Theragnostics, “Mariano Santo” Cosenza Hospital, 87100 Cosenza, Italy; f.calabria@aocs.it; 2Operative Medical Physics Unit, Cosenza Hospital, 87100 Cosenza, Italy; 3Department of Pharmacy, Health and Nutritional Sciences, University of Calabria, 87036 Rende, Italy; 4Complex Operative Oncology Unit, Annunziata Hospital Cosenza, 87100 Cosenza, Italy; 52nd Department of Radiology, University General Hospital “ATTIKON”, Medical School, National and Kapodistrian University of Athens, 12462 Athens, Greece; 6Department of Diagnostic Imaging, Oncologic Radiotherapy and Hematology, Fondazione Policlinico Universitario Agostino Gemelli IRCCS, 00100 Rome, Italy; 7Facoltà Di Medicina E Chirurgia, Università Cattolica del Sacro Cuore, 00100 Roma, Italy

**Keywords:** HIFU, LIFU, MRgFUS, oncology, thermal ablation

## Abstract

Magnetic Resonance-guided Focused Ultrasound (MRgFUS) represents a groundbreaking advancement in ultrasound-based cancer treatments. This innovative technology integrates high-intensity focused ultrasound (HIFU) or low-intensity focused ultrasound (LIFU) with a magnetic resonance imaging (MRI) system. In this paper, the physical principles of focused ultrasound, the basic technical aspects of MRgFUS, and its current and prospective oncologic applications will be briefly explored.

## 1. Introduction

High-intensity focused ultrasound (HIFU) is a non-invasive treatment that deposits energy inside the body without causing harm. The first preclinical publication dates back to 1942 when Lynn et al. [1] tested a “focused supersonic beam” in the animal brain. Since then, and for over eighty years, it has been gradually utilized as a reliable and effective technology for various medical applications in humans.

The primary effects of HIFU on the tissue include thermal heating and mechanical forces. Magnetic Resonance-guided Focused Ultrasound (MRgFUS) is an innovative and non-invasive technique that employs HIFU alongside magnetic resonance imaging (MRI) system guidance to target and deliver energy inside the lesion, resulting in considerable tissue heating that leads to necrosis in the target focal zone (Figure 1).

The development of MRgFUS devices continues to be an active area of research and clinical trials. This combined system offers two principal advantages: first, it provides highly accurate and comprehensive information regarding tumor localization, thereby facilitating three-dimensional treatment planning; second, it offers continuous thermal monitoring of the treatment area, aided by MR thermography. Implementing MRgFUS ablation enables exact tumor targeting, resulting in immediate temperature elevation within the specified zone and subsequent induction of cell death. Other non-thermal effects also contribute to tissue destruction.

Compared to other ablation technologies, this approach is characterized by its less invasive nature and the elimination of the need for complex image-guided interventional skills [2,3].

MRgFUS is an alternative treatment for various oncological, neurological, and musculoskeletal diseases. In oncology, it is used to treat primary and secondary bone lesions, as well as certain solid tumors [4].

This brief review aims to explain the physical principles behind focused ultrasound therapy, along with technical notes on MRgFUS and an overview of its established and emerging clinical applications that are in both early translational and clinical phases.

## 2. Physical Principles, Biological Effects, and Technical Notes

Medical therapeutic ultrasound is classified into two types based on intensity: high-intensity focused ultrasound (HIFU) and low-intensity focused ultrasound (LIFU). Both techniques utilize a concave transducer, lens, or phased array to focus ultrasonic waves into a precise tissue volume. The intensity is measured as the total power delivered per unit area in the focal region (W/cm^2^). Considering a train of sonic waves propagating in an absorbing medium with an attenuation coefficient μ, the intensity (I) of the ultrasound at depth x follows the following exponential law [5,6]:I(x) = I_0_ e^−μx^
where I_0_ is the intensity of the ultrasound beam at the point of generation (x = 0), while μ is the attenuation coefficient of the medium per unit path. Specifically, the attenuation coefficient of the incident beam intensity depends on the biological tissue and the ultrasound frequency according to a power law with the following form [5,6]:μ = a *f ^b^*
where *f* is the frequency, and the coefficients a and *b* are tissue-specific constants. It is important to underline that it is precisely this dependence of the attenuation coefficient, and therefore of each specific biological tissue, on the frequency that makes ultrasound particularly suitable for non-invasive and targeted therapy. However, it also presents significant challenges in optimizating the target destruction induced by HIFU. According to a power law, increasing the ultrasound frequency resulting in a higher attenuation coefficient. This implies that there will be greater heat deposition and a decrease in the penetration depth are obtained. Therefore, in a surgical procedure, selecting the appropriate ultrasound frequency is specific for each application and represents a compromise between the treatment depth and the desired heating rate at the target.

Frequencies ranging from 0.5 to 1 MHz have proven to be more effective for heat deposition during deep treatments, while frequencies as high as 8 MHz are convenient for more superficial treatments [7,8].

Current commercially available MRgFUS systems employ multi-element phased array ultrasonic transducers composed of a variable number (from several hundred to thousands) of individual piezoelectric ultrasonic elements. These transducers can be integrated into the MRI table or be dedicated and relocatable. As ultrasound waves travel through tissue, they are absorbed and can induce both thermal and non-thermal (mechanical) effects. The ultrasound waves are focused into a very small volume, the focal zone, which significantly increases the intensity of the sound waves, inducing energy release in the target tissue, defined as sonication. Within the focal zone, this high energy density raises the temperature to over 60 °C within seconds, leading to the denaturation of tissue proteins. Extensive denaturation of proteins has an immediate cytotoxic effect resulting in coagulative necrosis of tissue in the focal zone; additional lethal effects are associated with the loss of cell membrane integrity, mitochondrial dysfunction, and inhibition of DNA replication [9]. The resulting necrotic lesions are typically small and elliptical, measuring 50–300 mm^3^ in volume [10]. Combining multiple single lesions facilitates treating larger target volumes, such as solid tumors, possible without leaving gaps (Figure 2).

Adequate pauses between individual sonications are crucial to avoid tissue boiling and bubbles formation. These bubbles can reflect and distort the ultrasound field, potentially resulting in unpredictable lesion growth, insufficient treatment of the target volume, and unintended lesion formation in surrounding areas [10]. Another phenomenon requiring considersation when using HIFU is the mechanical effect of ultrasonic energy. FUS (focused ultrasound) can cause the oscillation of small gas bubbles trapped in the tissues, a process called *cavitation*. These bubbles undergo repeated cycles of rarefaction and compression. If the mechanical pressure continues to increase, the bubbles reach a threshold size where they collapse violently during the compression part of the cycle. This phenomenon is called inertial cavitation [11]. Cavitation only occurs only with high-intensity ultrasound waves, generating high pressures and temperatures, significant shear stress, and microflow jets of liquid that can damage cell walls. In a predominantly liquid, freely moving medium, the movement of the liquid creates microscopic flows contributing to cell apoptosis. The nuclei of the apoptotic cells ultimately self-destruct through deoxyribonucleic acid degradation by endonucleases [12].

The effectiveness of HIFU ablation largely stems from its ability to induce these mechanical effects, particularly cavitation, which is crucial for the procedure’s usefulness.

MRI offers detailed 3D planning in targeting specific tissues, demonstrating high tumor detection sensitivity and excellent anatomical resolution. Additionally, it allows for the pre-definition of sonication position, size, and physical characteristics (such as energy, frequency, and duration) [13].

LIFU uses ultrasonic sound waves, often pulsed in a sinusoidal waveform, to propagate through bone and tissue. For the human skull, the range used is typically 250 kHz to 650 kHz. Unlike HIFU, LIFU results in low intensities (0.125–3 W/cm^2^) of focused ultrasound and causes low-temperature mechanical agitation [14,15].

Real-time imaging during therapeutic procedures is crucial for ensuring both safety and treatment effectiveness. Moreover, MR-thermometry allows the measurement of the thermal doses, also providing a superimposed representation of the anatomical area where temperatures reach cytotoxic levels. During the treatment, MR-thermometry controls energy deposition with a temperature accuracy of ±1 °C, a spatial resolution of 1 mm, and a temporal resolution of 1 s. This precision is achievable because many MRI parameters, including T1 and T2 relaxation times, proton resonance frequency, and magnetization transfer coefficients, exhibit temperature dependencies that can be utilized. In proton-resonance frequency shift thermometry, temperature-dependent phase changes in gradient-recalled echo pulse sequences are used to determine temperature variations [3,16,17].

At the end of the procedure, the success of the treatment is assessed by T1-weighted MR images enhanced with gadolinium-based contrast material, where non-enhancing regions indicate the Non-Perfused Volume (NPV), an MR biomarker of clinical efficacy [18].

MRI, utilized for disease detection and diagnosis, can also play a valuable role in the real-time assessment of precise energy delivery inside tumor lesions and, consequently, in the evaluation of response to therapy, allowing the assessment of the completeness of the ablation during follow-up [19].

## 3. An Overview of Current and Prospective Applications in Oncology

MRgFUS is a valuable incisionless option in the multidisciplinary management of primary and secondary malignant tumors as well as painful bone metastases. It also attracts significant interest from the cancer community to explore its therapeutic potential in the various areas of oncology.

The Table 1 enumerates the active and recruiting clinical trials associated with certain applications of MRgFUS, derived from the ClinicalTrial.gov database (accessed 15 December 2024), which will be explained in the following paragraphs.

### 3.1. Bone Metastases

Bone metastases play a significant role in causing cancer-related pain, which can greatly impact quality of life. Radiotherapy (RT) and analgesics represent the standard of care for the management of localized metastatic pain; however, approximately two-thirds of patients are known to continue to experiencing residual pain after treatment [20]. MRgFUS is a non-invasive treatment that utilizes focused ultrasound waves to destroy the pain-transmitting periosteal nerves at the bone surface near the tumor. This approach delivers rapid and lasting relief for patients suffering from *painful bone metastases.*

HIFU is believed to alleviate pain through thermal denervation of the bone and periosteum; however, the volumetric reduction of metastases may also play a role [21].

The acoustic absorption and low thermal conductivity of cortical bone limit the diffusion of focused ultrasound energy to its surface. Focusing acoustic energy on the intact surface of cortical bone rapidly raises the temperature, causing critical thermal damage to the adjacent periosteum for pain relief. In the ablation of bone lesions, there are two scenarios. With intact cortical bone, the equipment set includes lower-frequency modulated protocols that reach the medullary and subcortical lesion; when the cortex is destroyed, as there are no absorption barriers, a high-frequency protocol with multiple ablative spots inside the lesion will be used [22].

MRgFUS effectively delivers complete or partial pain relief in approximately 79% (95% CI: 73–83%) of patients. Furthermore, it maintains low-grade and high-grade treatment-related adverse event rates below 6% and 1%, respectively [23,24].

Patients typically experience pain relief within 3 days to 2 weeks after the treatment, many reducing or stopping their pain medication. Bony targets are localized (painful) non-spinal lesions (apart from the posterior elements below the level of the conus medullaris) or non-cranial lesions, which can be identified by radiological imaging. The safety distance from the skin and the main nerve bundles is ≥1 cm; bone injuries should not require surgical stabilization, with low fracture risk (Mirel score ≤ 7). There should be no interposition of non-target bone, hollow viscera with airborne contents, or extensive scars along the trajectory of the ultrasound beam [25]. MRgFUS is a valuable opportunity in the management of painful bone metastasis, particularly in patients who have previously not responded to RT [26].

Additionally, MRgFUS can be recommended as an alternative to external beam radiation therapy or as a follow-up for patients with skeletal oligometastases (1–5 metastatic bone lesions).

Devices are accessible, representing a highly effective therapy that can provide local tumor control (LTC) in 84% (95% CI: 66–97%) of lesions treated, which is superior to other percutaneous ablation procedures that achieved 65% (95% CI: 51–78%) of LTC [2].

Compared to RT, MRgFUS has the advantage of not having dose limits and allows for more treatment sessions, making it an effective option, especially for non-radiosensitive tumors or when a dose limit for adjacent radiosensitive organs is conceivable. Furthermore, a significant pain relief can typically be achieved in a single treatment session [27].

Two clinical trials are currently recruiting participants. One (NCT05167669) evaluates the effectiveness of mild adjuvant hyperthermia combined with external beam radiation therapy (EBRT) in managing bone metastasis pain. Another (NCT04307914) assesses the effectiveness and cost-effectiveness of MRgFUS (either alone or in combination with EBRT) compared to EBRT alone as the standard of care, for palliative treatment options for cancer-induced bone pain.

### 3.2. Prostate Cancer

MRgFUS can effectively treat patients with intermediate-risk *prostate cancer* who wish to avoid radical whole-gland treatment, with a low rate of genitourinary adverse effects [28]. Two types of devices have been developed and tested: transrectal and transurethral MRgFUS systems.

In a prospective phase II trial by Ghai et al. [29], 41 of 44 men (93%; 95% CI: 82–98%) had no residual disease at the treatment site with a transrectal MRgFUS system, and only one man (2%; 95% CI: 0.4–11.8%) experienced severe pelvic pain that persisted and reported a grade 3 adverse event at 5 months following treatment. This therapy is suitable for individuals with unilateral, MRI-visible, primary, intermediate-risk, previously untreated prostate adenocarcinoma, characterized by a prostate-specific antigen level of ≤20 ng/mL, Grade Group 2 or 3, and tumor classification of ≤T2, all confirmed via fusion biopsy. After 2 years, 39 of the 43 participants (91%) exhibited no evidence of clinically significant prostate cancer at the treatment site, and 36 of the 43 participants (84%) were found to have no cancer present in the entire gland, based on the results from multiparametric MRI as well as targeted and systematic biopsy follow-up [30]. In a Phase I trial by Chin et al. [31], the transurethral MRgFUS system (MRI-TUSLA) showed that men with biopsy-proven low-risk (80%) and intermediate-risk (20%) prostate cancer had a median PSA reduction of 87% at one month, stabilizing at 0.8 ng/mL by twelve months. There was also a 61% decrease in total cancer length, with clinically significant disease in 9 of 29 patients (31%; 95% CI: 15–51%) and any disease in 16 of 29 patients (55%; 95% CI: 36–74%).

In this trial, the MRgFUS treatment included a minimum margin of 3–5 mm in the target area, and in some cases, up to 10 mm, beyond the tumor visible on the magnetic resonance imaging was included in the treatment plan. This procedural aspect increases the likelihood of treating the entire volume of the histological tumor during focal ablative therapy. Furthermore, using thermographic maps overlaid on anatomical MR images during treatment allowed for monitoring of the effective ablation, and the effects on urinary control, erectile function, pain, and morbidity were minimal.

### 3.3. Breast Cancer

MRgFUS appears to be a non-invasive treatment for localized, clinically non-palpable *early-stage breast cancer*, potentially replacing lumpectomy [32]. HIFU breast cancer therapy offers, various other benefits, including the preservation of tissue integrity, absence of scars, and slight changes in breast shape with good cosmetic results.

Merckel et al. [33] report their initial experiences with MRgFUS ablation for breast cancer using a dedicated breast platform, finding it safe and resulting in confirmed tumor necrosis. The dedicated MRgFUS platform for the breast is a working model with the breast targeted in the cup of the breast filled with water, surrounded by transducers distributed along a circular arc of 270°. During treatment, patients are positioned prone on the table with the targeted breast inside the cup [34]. In a prospective study of Furusawa et al. [35], thirty women with invasive breast cancer (T1–2, N0–2, M0) were treated using MRgFUS, followed by excision or mastectomy. Tumor histology revealed that the average necrosis of the targeted breast tumors was 96.9% of the tumor volume. Treatment was generally well tolerated, with few adverse effects; while only one patient suffered a third-degree burn due to user error. Pathology from a lumpectomy indicated that residual tumors in two patients were at the margins of the treatment area, underscoring the need for a 5-mm safety margin in the prescribed treatment region.

Patients with large T2 tumors, however, are generally not appropriate candidates for treatment with minimally invasive ablation techniques such as the HIFU. Furthermore, it is important to consider how the location of the tumor within the breast can impact treatment options and outcomes. If the tumor is close to the skin, there is a risk of burns or leaving tumor cells if the overlying tissue is not treated properly.

The phase I clinical trial (NCT05291507), currently in the recruitment phase, aims to evaluate the efficacy and safety of an MRgFUS system dedicated to breast lesion ablation (a custom breast MR coil and an ultrasound transducer that can remotely steer a beam to target nearly any ablation zone in the breast, along with a customized comfort table for patients to lie face-down during the procedure), study is designed as an analysis of partial ablation (50%) of the tumor on the operative after standard surgical resection.

### 3.4. Abdominal Cancers

MRgFUS is a feasible and repeatable ablative technique for patients with unresectable and device-accessible *hepatic and pancreatic lesions* [36]. Preliminary results from Anzidei et al. [37] in one patient with unresectable hepatocellular carcinoma, distant from intrahepatic vessels, and two patients with pancreatic body adenocarcinoma are encouraging, particularly regarding the pain relief response and the palliation of local tumor growth. MRgFUS effectively controls tumors and alleviates symptoms in pancreatic adenocarcinoma by damaging the celiac plexus. In hepatocellular carcinoma, lesion ablation is confirmed through imaging and histopathology. Once fast organ tracking technology becomes available, it has the potential to serve as an alternative to surgical resection for *malignant renal tumors* [38].

### 3.5. Targeted Drug Delivery

A promising application of MRgFUS is targeted drug release through the systemic administration of nanocarriers sensitive to mechanical forces and/or sensitive temperature [39]. Anticancer drugs (e.g., Doxorubicin) can be encapsulated in gas-filled bubbles or thermosensitive liposomes attached to nanoparticles. These are released locally by rupturing the bubbles during sonication of the target area (Figure 3).

A clinical trial in the recruitment phase (NCT04791228) aims to evaluate the use of a Lyso-thermosensitive Liposomal Doxorubicin (LTLD) formulation that releases the drug when exposed to hyperthermal conditions (40–45 °C) mediated by MRgFUS. The approach involves the continuous maintenance of the target in mild hyperthermia after the infusion of LTLD. This procedure is combined with ablation therapy (>55 °C) in targeted areas of the tumor, where possible and safe, to significantly enhance chemotherapy with minimal additional adverse effects, thereby improving local control in subjects with refractory/relapsed solid tumors.

### 3.6. Immunological Effects

There is also growing interest in combining immunotherapy with ablative strategies such as MRgFUS due to mechanical changes in the tumor microenvironment and inflammatory-mediated changes in immune phenotypes [40,41]. Focused ultrasound can stimulate immune responses in tumors by releasing tumor-associated antigens. This process can lead to T cell-specific responses, an increase in tumor-infiltrating lymphocytes, key immune cells in both humoral and cellular antitumor immune responses, and antigen-presenting cells within the tumor microenvironment (Figure 4). Additionally, it can alter the immune context of the tumor, induce the abscopal effect, and reverse T cell anergy and tolerance [42,43,44].

Given the above considerations, the use of lymphocyte count and the lymphocyte-to-monocyte ratio as predictors of pre-procedure outcomes could be considered relevant in the treatment of solid tumors, as already demonstrated for patients with liver metastases from colorectal carcinoma treated with percutaneous ablative techniques [45].

A clinical trial (NCT04123535) began with 20 adult participants to examine immune responses in patients receiving MRgFUS before surgical resection or biopsy of the undifferentiated pleomorphic sarcoma. The study will use serological flow cytometry for T cells and natural killer cells, multiplex histochemistry of resected tumor samples, and RNA sequencing to assess the effects of the immune response.

### 3.7. Neuro-Oncology

Transcranial MRgFUS (tcMRgFUS) has mainly been used for movement disorders [46]. Since the FDA approved unilateral thalamotomy for drug-refractory essential tremor in 2016, its applications have also expanded into neurosurgical adult and pediatric oncology. The first successful treatment using HIFU thermal ablation in a Glioblastoma Multiforme was conducted by Coluccia et al. in 2014 [47]. Recent studies have examined low-intensity pulsatile focused ultrasound (LIFU) for temporarily enhancing the blood-brain barrier (BBB) permeability [48]. Unlike HIFU, which can cause permanent brain damage, LIFU safely disrupts the tight junctions between endothelial cells. This allows the barrier to open briefly for several hours. Combining this technique with the administration of exogenous microbubbles (i.e., ultrasound contrast) could improve chemotherapy trans-BBB delivery for the targeted treatment of primary and secondary brain tumors, maximizing local concentration and reducing systemic toxicity [49,50]. Several studies have indicated that tcMRgFUS can effectively stimulate innate and acquired immune responses [51]. A Phase 0 single-center, first-in-human, clinical trial (NCT04559685) was started using MRgFUS combined with intravenous aminolevulinic acid, a promising sonosensitizer that, once selectively absorbed by cancer cells is metabolized and accumulated as protoporphyrin IX, leading to apoptosis [52], to assess safety and efficacy in more than 45 participants with recurrent High-Grade Glioma.

Table 2 summarizes the principal research conducted on MRgFUS, categorized by its respective fields of application and document type.

## 4. Conclusions

The effectiveness of the MRgFUS technique is closely tied to identifying lesions accessible by the device and the absence of contraindications to MRI. MRgFUS has several advantages, such as the repeatability of the procedure if necessary and a favorable safety profile, with no dose limit reported for focused ultrasound energy. Given the clinical circumstances of patients, MRgFUS can be considered a viable treatment option, either on its own or in combination with other therapies to cure the disease.

In treating bone metastases, this procedure is an established palliative approach, demonstrating evidence of therapeutic efficacy during the oligometastatic phase. Its potential is particularly noteworthy for prostate and breast tumors in the early stages of the disease, allowing for aesthetic and functional respect for sex-specific organs.

Evidence for the use of this procedure in solid abdominal tumors is still limited; however, with the advent of respiratory movement control systems, we can expect this knowledge to expand. The emerging applications in neuro-oncology are especially valuable, as they encompass two innovative aspects of the procedure: targeted delivery of chemotherapy with enhanced local diffusion and also immunomodulation.

At present, the term “*Evolving Nonsurgical Precision Ablation*” appears to be an appropriate descriptor for specific MRgFUS applications in oncology. Combining this approach with other interventions and drug delivery may enhance their effectiveness and therapeutic outcomes. Advances in platforms and devices will soon enable the expansion of anti-cancer MRgFUS applications.

## Figures and Tables

**Figure 1 cancers-17-00036-f001:**
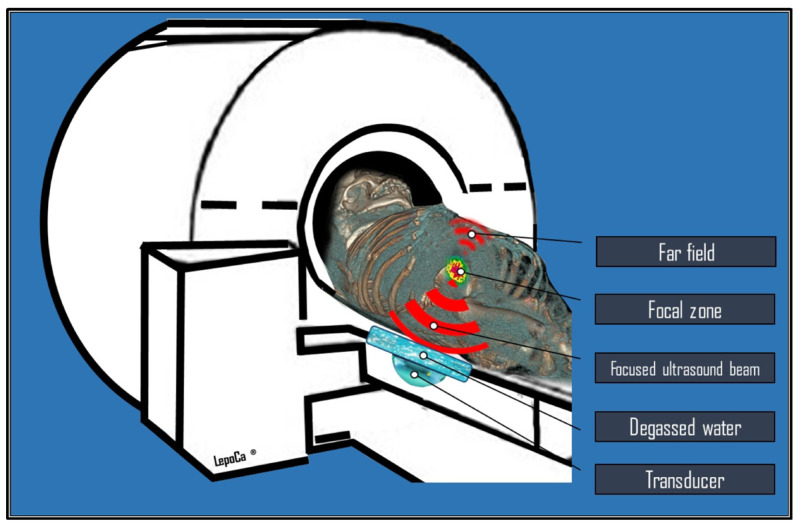
The diagram illustrates an MRI scanner equipped with a high-intensity focused ultrasound (HIFU) transducer on the patient table. It showcases the configuration of the MR-guided Focused Ultrasound (MRgFUS) system, which includes an integrated focused ultrasound unit within the MRI bed. Patients are positioned carefully to ensure that the area of interest aligns directly above the FUS transducer. The concave transducer, submerged in degassed water and using a coupling gel pad, effectively transmits acoustic waves through the patient’s body. This multi-element ultrasonic transducer focuses ultrasound waves on the targeted area (focal zone), generating heat that leads to precise tissue ablation via necrosis. This advanced procedure is performed while the patient is inside the bore-MRI, always adhering to strict treatment planning protocols.

**Figure 2 cancers-17-00036-f002:**
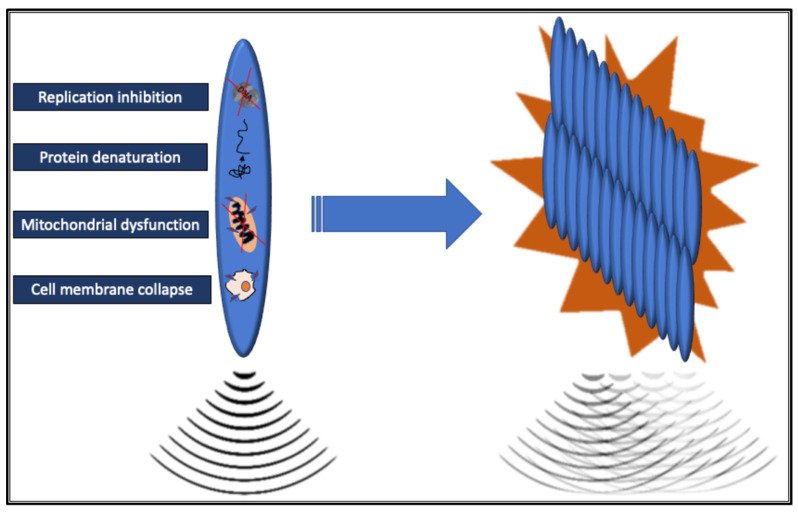
Schematic drawing of the sonication and thermal cytotoxic effects of the HIFU in the focal zone. The necrotic post-ablative lesions are elliptical; multiple sonications without gaps are necessary to target the entire lesion and achieve radical tumor ablation.

**Figure 3 cancers-17-00036-f003:**
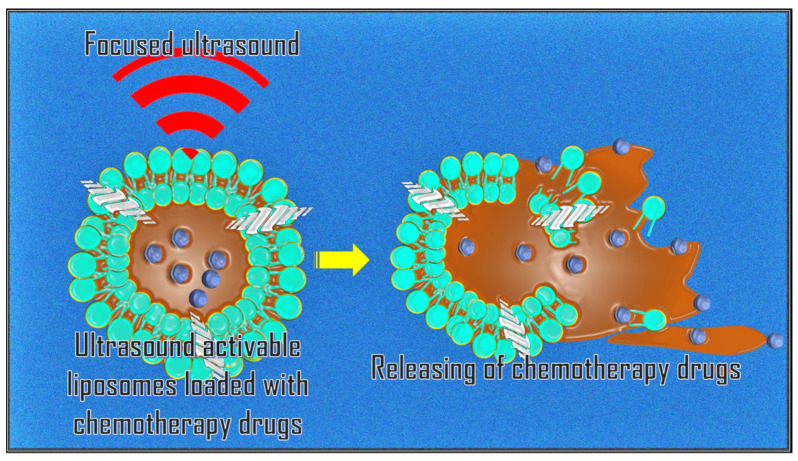
The thermo-mechanical effect of HIFU can be exploited to improve drug distribution and absorption by promoting the release of anticancer molecules encapsulated in a carrier (liposome) within the target tumor site.

**Figure 4 cancers-17-00036-f004:**
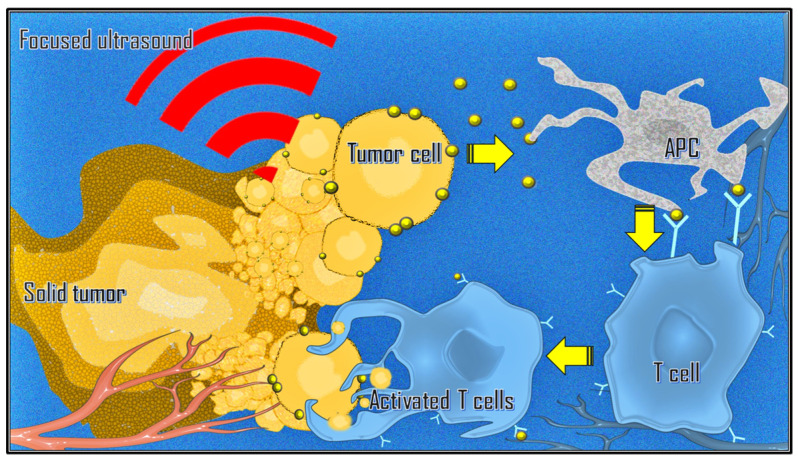
HIFU ablation can induce an immune response. The generation of tumor debris in situ increases the circulation of tumor-associated antigens. This activates the immune response mediated by the interaction between antigen-presenting cells (APCs) and T lymphocytes (T cells) and will target cancer cells that expose that specific antigen. The activated T lymphocytes can infiltrate the tumor site and attack tumor cells by passing through the systemic circulation.

**Table 1 cancers-17-00036-t001:** Ongoing clinical trials (active or in the recruitment phase) on applications of MRgFUS in the oncology field. Data extracted from the ClinicalTrial.gov database, accessed 15 December 2024.

NCT Number	Study Title	Conditions	Status
NCT05291507	Feasibility Evaluation of the Muse Magnetic Resonance Guided Focused Ultrasound	Breast Cancer	Recruiting
NCT05167669	Pain Relief in Symptomatic Bone Metastases with Adjuvant Hyperthermia MR Guided	Bone Metastases and Pain	Recruiting
NCT04791228	A Pilot Study of Thermodox and MR-HIFU for Treatment of Relapsed Solid Tumors	Solid Tumors	Recruiting
NCT04559685	Study of Sonodynamic Therapy in Participants With Recurrent High-Grade Glioma	High Grade Glioma	Recruiting
NCT04307914	Focused Ultrasound and RadioTHERapy for Noninvasive Palliative Pain Treatment in Patients With Bone Metastases	Cancer Induced Bone Pain	Recruiting
NCT04123535	Focused Ultrasound to Promote Immune Responses for Undifferentiated Pleomorphic Sarcoma	Undifferentiated Pleomorphic Sarcoma	Recruiting
NCT03028246	A Feasibility Safety Study of Benign Centrally-Located Intracranial Tumors in Pediatric and Young Adult Subjects	Benign Centrally-Located Intracranial Tumors	Recruiting
NCT02076906	MR-guided High Intensity Focused Ultrasound (HIFU) on Pediatric Solid Tumors	Relapsed and Refractory Pediatric Solid Tumors	Active

**Table 2 cancers-17-00036-t002:** This table summarizes some of the main publications on MRgFUS applications in specific oncology fields, with particular reference to clinical reviews, meta-analyses, and trials from the last twenty years.

	1st Author, Year	Title	Document Type
Bone Metastases	Leporace, 2024 [2]	Magnetic resonance-guided focused ultrasound versus percutaneous thermal ablation in local control of bone oligometastases: a systematic review and meta-analysis	Meta-Analysis
	Napoli, 2013 [22]	MR imaging-guided focused ultrasound for treatment of bone metastasis	Review
	Baal, 2021 [23]	Efficacy and safety of magnetic resonance-guided focused ultrasound for the treatment of painful bone metastases: a systematic review and meta-analysis	Meta-Analysis
	McGill, 2024 [24]	Update on musculoskeletal applications of magnetic resonance-guided focused ultrasound	Review
	Yeo, 2022 [25]	High Intensity Focused Ultrasound for Treatment of Bone Malignancies-20 Years of History	Review
	Hurwitz, 2014 [26]	Magnetic resonance-guided focused ultrasound for patients with painful bone metastases: phase III trial results	Clinical Trial
	Huisman, 2015 [27]	International consensus on use of focused ultrasound for painful bone metastases: Current status and future directions	Review
Prostate Cancer	You, 2016 [28]	Focal therapy using magnetic resonance image-guided focused ultrasound in patients with localized prostate cancer	Research Article
	Ghai, 2021 [29]	MRI-guided Focused Ultrasound Ablation for Localized Intermediate-Risk Prostate Cancer: Early Results of a Phase II Trial	Clinical Trial
	Ghai, 2024 [30]	MRI-guided Focused Ultrasound Focal Therapy for Intermediate-Risk Prostate Cancer: Final Results from a 2-year Phase II Clinical Trial	Clinical Trial
	Chin, 2016 [31]	Magnetic Resonance Imaging-Guided Transurethral Ultrasound Ablation of Prostate Tissue in Patients with Localized Prostate Cancer: A Prospective Phase 1 Clinical Trial	Clinical Trial
Breast Cancer	Matsutani, 2020 [32]	Innovative use of magnetic resonance imaging-guided focused ultrasound surgery for non-invasive breast cancer: a report of two cases	Research Article
	Merckel, 2016 [33]	First clinical experience with a dedicated MRI-guided high-intensity focused ultrasound system for breast cancer ablation	Research Article
	Merckel, 2013 [34]	MR-guided high-intensity focused ultrasound ablation of breast cancer with a dedicated breast platform	Review
	Furusawa, 2006 [35]	Magnetic resonance-guided focused ultrasound surgery of breast cancer: reliability and effectiveness	Clinical Trial
Abdominal Cancers	Orsi, 2010 [36]	High intensity focused ultrasound ablation: a new therapeutic option for solid tumors	Review
	Anzidei, 2014 [37]	Magnetic resonance-guided focused ultrasound ablation in abdominal moving organs: a feasibility study in selected cases of pancreatic and liver cancer	Research Article
	Saeed, 2016 [38]	Renal ablation using magnetic resonance-guided high intensity focused ultrasound: Magnetic resonance imaging and histopathology assessment	Research Article
Targeting Drugs Delivery	Thanou, 2013 [39]	MRI-Guided Focused Ultrasound as a New Method of Drug Delivery	Review Article
Immunological Effects	Silvestrini, 2017 [40]	Priming is key to effective incorporation of image-guided thermal ablation into immunotherapy protocols	Research Article
	Joiner, 2020 [41]	Focused Ultrasound for Immunomodulation of the Tumor Microenvironment	Review
	Lu, 2009 [42]	Increased infiltration of activated tumor-infiltrating lymphocyte safter high intensity focused ultrasound ablation of human breast cancer	Randomized Controlled Trial
	Xu, 2009 [43]	Activation of tumor-infiltrating antigen presenting cells by high intensity focused ultrasound ablation of human breast cancer	Randomized Controlled Trial
	Xia, 2012 [44]	High-intensity focused ultrasound tumor ablation activates autologous tumor-specific cytotoxic T lymphocytes	Research Article
Neuro-Oncology	Coluccia, 2014 [47]	First noninvasive thermal ablation of a brain tumor with MR-guided focused ultrasound	Case Study
	McMahon, 2019 [48]	Evaluating the safety profile of focused ultrasound and microbubble-mediated treatments to increase blood-brain barrier permeability	Review
	Beccaria, 2020 [49]	Blood-brain barrier disruption with low-intensity pulsed ultrasound for the treatment of pediatric brain tumors: a review and perspectives	Review
	Grasso, 2023 [50]	MR-guided focused ultrasound-induced blood-brain barrier opening for brain metastasis: a review	Review

## Data Availability

The contributions presented in this study are included in the article. Further inquiries can be directed to the corresponding author.

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
