# Peer review of "The Thermal Ablation with MRgFUS: From Physics to Oncological Applications"

_cancers, 2024, doi:10.3390/cancers17010036_

Round 1
Reviewer 1 Report
Comments and Suggestions for Authors
The review is interesting and well written. My comments:
1) The authors could add a table summarizing the mains studies in the field
2) THe authors could search Trialgov and the main databases to check if some RCTs are currently ongoing in this field
3) Is there some immunomodulating effects of this treatment as already demonstrated with other ablative techniques? The authors should comment that some potential immune-related prognostic factors have been demonstrated with thermal techniques for example in the liver (in this regard cite the paper PMID: 27122671) and the same could be (or could not be.....) with this technique
Author Response
Thank you for appreciating our work and for your time.
- Two tables were added to the paper, respectively summarizing published work on the topic and current ongoing clinical trials, as by you kindly suggested;
- The immunomodulating effects of the procedure were adequately described; a reference was added; thank you for the suggestion;
We hope you may appreciate our efforts and consider the paper for possible publication in the Journal.
Kindest regards.
Reviewer 2 Report
Comments and Suggestions for Authors
Please check the attached file.

Comments on the Quality of English Language
Please check comments for authors
Author Response
Thank you for your kind attention and positive comments.
- in the abstract, more was mentioned about Oncology and the relevance of the technique explained, as you kindly noticed.
- The introduction was expanded, in order to more adequately describe the targeted illness.
- Comments and information were added to the legend of the Figure 1.
- The aim of the review was more clearly explained in the Introduction section.
- We effectively improved the contents of the paper, expanding the discussion section and adding table 1, focused on the main papers on the topic. Concerning the new trends, we also cited and discussed new ongoing clinical trials on the MRgFUS in different clinical settings. We hope this can improve the panorama of this review, as you kindly requested.
- The conclusions section was improved.
- Figure legends were improved. “Permissions and Right of the Author, Editorial, year” were released to the editorial office. Thank you for the suggestion.
- Specific targeting pathways were added to the ”drug delivery” section.
Round 2
Reviewer 1 Report
Comments and Suggestions for Authors
The manuscript is OK now
Reviewer 2 Report
Comments and Suggestions for Authors
Thank you very much for your responses and modifications. Now, the manuscript is accepted for further editorial processing and publication.